# Prevalence and Correlates of Sarcopenia among Elderly CKD Outpatients on Tertiary Care

**DOI:** 10.3390/nu10121951

**Published:** 2018-12-10

**Authors:** Claudia D’Alessandro, Giorgina Barbara Piccoli, Massimiliano Barsotti, Serena Tassi, Domenico Giannese, Riccardo Morganti, Adamasco Cupisti

**Affiliations:** 1Department of Clinical and Experimental Medicine, University of Pisa, 56126 Pisa, Italy; dalessandroclaudia@gmail.com (C.D.); serenatassi@msn.com (S.T.); adamasco.cupisti@med.unipi.it (A.C.); 2Department of Clinical and Biological Sciences, University of Torino, 10124 Torino, Italy; 3Nephrologie, CH Le Mans, 72037 Le Mans, France; 4Nephrology, Transplant and Dialysis Unit, AOUP, 35233 Pisa, Italy; m.barsotti63@gmail.com (M.B.); d.giannese@ao-pisa.toscana.it (D.G.); 5Section of Statistics, AOUP, 56100 Pisa, Italy; morganti@ao-pisa.toscana.it

**Keywords:** CKD, aging, sarcopenia, physical performance, functional capacity, protein restriction

## Abstract

Background: Sarcopenia is a widespread concern in chronic kidney disease (CKD) as well in elderly patients and is one of the main reasons why low-protein diets for this population are controversial. The aim of this study was to assess the prevalence and correlates of sarcopenia among elderly male patients affected by CKD followed up in an outpatient nephrology clinic, where moderate protein restriction (0.6–0.8 g/Kg/day) is routinely recommended to patients in CKD stage 3b-5 not on dialysis. Methods: This observational study included 80 clinically-stable male out-patients aged >60, affected by stage 3b-4 CKD. Forty patients aged ≥75 (older seniors) were compared to the other forty patients aged 60–74 (younger seniors). All patients underwent a comprehensive nutritional and functional assessment. Results: Older seniors showed lower serum albumin, hand-grip strength, body mass index (BMI), skeletal muscle mass, and resting energy expenditure. Protein intake was significantly lower in older seniors whereas energy intake was similar. Average daily physical activity was lower in the older seniors than in the younger ones. Sarcopenia was more prevalent in older than in younger seniors. Among older seniors, sarcopenic and non-sarcopenic ones differed in age and performance on the Six-Minute Walk test, whereas the estimated glomerular filtration rate (eGFR), biochemistry, dietary protein, and energy intakes were similar. Conclusions: Older senior CKD male patients have lower muscle mass, muscle strength, and physical capacity and activity levels, with a higher prevalence of sarcopenia than younger patients. This occurs at the same residual renal function and metabolic profile and protein intake. Energy intake was at the target in both subgroups. In this CKD cohort, sarcopenia was associated with age and physical capacity, but not with eGFR or dietary intakes.

## 1. Introduction

In the context of the overall prevalence of chronic kidney disease (CKD), averaged at roughly 10–15% in most highly resourced countries, CKD prevalence increases with age, and affects up to one-third of individuals aged over 60. In this “older” population, CKD is often associated with additional comorbid conditions, high cardiovascular risk, and increasing prevalence of frailty, disability, and malnutrition [1,2]. Furthermore, abnormalities in body composition, nutritional status, physical activity, and performance are all associated with poor quality of life, and increased risk of protein-energy wasting, morbidity, and mortality [3].

Nutritional management in older adults is extremely complex as many factors, such as chewing problems, lack of appetite, comorbidities, and depression, may affect their dietary habits. Elderly people often prefer sweet foods, probably for psychological reasons, and because store-bought sweets require little or no preparation [4].

The correct management of energy demand is mandatory to prevent the onset of protein-energy wasting (PEW) especially when patients are on low-protein regimes. Independent of protein restrictions, the prevalence of PEW in non-dialyzed patients ranges from 6 to 10% in people with mild to moderate CKD (depending on the method used to assess body composition) and increases to 20–40% in dialyzed patients. The prevalence is higher in elderly CKD patients, as reported in more than half of elderly dialysis individuals: This condition leads to an increase in protein catabolism, muscle-mass reduction, inflammatory cytokine production, poor appetite, and poor physical performance. In addition, PEW is a well-known unfavorable prognostic factor [5].

The scenario is even more complex in obese patients. In fact, obesity is also quite prevalent in CKD patients, contributes to physical limitations in older adults, and is an independent risk factor for CKD and cardiovascular disease [6,7].

In CKD patients, most of the current guidelines recommend an energy intake of 30 Kcal/kg per ideal body weight (IBW) per day for subjects over 60 years of age, compared to 35 Kcal/kg IBW/day in those under the age of 60. Normal to reduced intakes of protein (0.8–0.6 g/kg/day), sodium (100 mmol/day), and phosphorus (800–600 mg/day) are also recommended as means of correcting metabolic and nutritional abnormalities and preventing protein-energy wasting (PEW) [8]. These data do not take into account the different needs of obese patients, and may not be adequate for “very old” patients.

Diet is a complex issue: restriction of protein intake represents a chance to stabilize kidney function in selected patients with advanced CKD; however, a higher dietary protein intake is generally recommended for older people, given their impaired protein utilization capacity, but no recommendation has been established for older CKD patients [9].

Finally, a further concern is the high prevalence of sedentary lifestyle that characterizes both CKD patients and elderly individuals. Physical activity declines with age and older adults are the least active age group in the general population; inactivity is also associated with mobility limitations, chronic diseases, and obesity [10]. Sarcopenia, the term currently used to indicate reduced muscle mass and muscle function, is associated with reduced physical ability and mortality. Chronic kidney disease is included among the causes of sarcopenia and sarcopenia has been shown to be associated with increased mortality in patients with CKD [11,12,13,14]. Males with CKD seem to be more prone to sarcopenia than women with CKD [15]. The criteria for diagnosing sarcopenia have only recently been defined in a consensus statement by the European Working Group on Sarcopenia in Older People [15]. The consensus recommendations include the use of at least one reliable technique for muscle-mass assessment (dual-energy X-ray absorptiometry, computed tomography, or magnetic resonance imaging) which are however not always available in clinical practice. As a valid alternative, they propose bio-impedance analysis (BIA) as a user-friendly, non-invasive and readily available method [16].

The aim of our study was to evaluate the prevalence and correlates of sarcopenia, assessed using BIA, in a cohort of older male patients with advanced CKD.

## 2. Subjects and Methods

This study includes a convenience sample of 80 clinically-stable male out-patients over the age of 60, affected by stage 3b-4 CKD, being followed in our CKD outpatient clinic. The forty patients aged 75 or over (older seniors) were compared to the other 40 patients aged 60–74 (younger seniors). All patients signed an informed consent form for the use of their anonymous data.

All patients received standard care including dietary support, according to their residual kidney function; all underwent a comprehensive nutritional and functional assessment including biochemistry, anthropometry, bio-impedance, dietary recall, physical activity, and performance testing, as specified below.

Renal survival and patient survival data were added to contextualize the study population, with regard to the current literature.

### 2.1. Biochemistry

Biochemical tests included serum blood urea nitrogen (BUN), creatinine, albumin, phosphorus, calcium, bicarbonate, parathyroid hormone (PTH), hemoglobin, and hematocrit, performed using standard laboratory methods. Glomerular filtration rate (eGFR) was calculated using the CKD-EPI formula [17]. Urinary sodium, phosphate, and urea were measured on 24 h urine samples. Protein catabolic rate was calculated by the Maroni–Mitch formula as a surrogate of dietary protein intake.

### 2.2. Anthropometry

Body weight was assessed on a mechanical scale with the patient wearing light clothes and no shoes. Height was measured with a stadiometer. Body mass index (BMI) was calculated as body weight (kg)/height^2^ (m^2^). Other measures included non-dominant middle arm circumference (MAC), waist and hip circumferences, and triceps skinfold thickness. Triceps muscle circumference (middle arm muscle circumference, MAMC) and area (middle arm muscle area, MAMA) were also calculated.

Body composition was determined using a Bioelectrical Impedance Analyzer (BIA/STA, Akern, Florence, Italy) with a distal, tetrapolar technique, delivering an excitation current at 50 kHz.

BIA gives two bioelectric parameters: body resistance (R) and reactance (Xc), and the impedance vector (Z) is a combination of R and Xc across tissues. The arc tangent of Xc/R is called phase angle (PA), which is a derived measure obtained from the relation between the direct measures of resistance and reactance reflecting hydration status and soft tissue cellular mass. Reduced phase angle value reflects increased extra- to intra-cellular water ratio as well as reduced body cell mass. Body cell mass (BCM) and skeletal muscle mass (SM) were derived from bio-impedance analysis and body cell mass index (BCMI) and skeletal mass index (SMI) were consequently calculated [18].

### 2.3. Dietary Nutrient Intake

The Council on Nutrition Appetite Questionnaire (CNAQ) was self-administered [19].

Current daily energy intake was assessed using a 3-day dietary journal: to improve data quality, the journal was completed at home after a skilled renal dietitian had provided all the information necessary for its correct compilation. Once completed, the dietitian interviewed patients to verify the correct and complete description of their dietary habits. The amount of food consumed was identified by weighing or through the use of photographic images of portions (photographic atlas, real size pictures), that the dietitian showed patients during the interviews.

Macronutrient intake data was obtained using the updated Italian Food Composition Tables [20].

The 3-day dietary record was conducted on the basis of widely accepted and applied rules [21]. Food record analysis represents a valid and simple tool for nutrients and energy estimation but it is well known that potential errors, most often underestimation, may occur. We tried to limit this bias by collecting an interview at the time of delivery of the 3-day food journal, in order to verify, as described in the method section, the sizes of the portion, the type of food and to complete missing data. Recording a food journal for a longer period is probably more appropriate but also at a higher risk of underreporting [22], especially in older people who easily get tired answering too many questions.

Furthermore, we preferred the 3-day food record to the food frequency questionnaire since the latter is more appropriate for a qualitative rather than quantitative assessment of food habits, and, in this specific case, we felt that quantification of energy and proteins was of particular importance; moreover, the food frequency questionnaire is generally applied in study population rather than small cohorts [23].

### 2.4. Dietary Counseling

Patients received individualized dietary counseling whose aim was to obtain favorable metabolic changes, and correct or prevent signs and symptoms of renal insufficiency. Recommendations were based on the main principles of nutritional therapy in CKD, namely adequate energy intake, normal to moderately reduced protein intake, controlled sodium, and phosphate intake. Particular attention was given to the socio-economic condition of patients and their families to improve dietary adherence.

### 2.5. Resting Energy Expenditure (REE)

Resting energy expenditure (REE) was measured by indirect calorimetry with a hand-held desktop calorimeter (Fitmate GS, Cosmed, Rome, Italy) using the dilution technique. The Fitmate calorimeter’s accuracy has been validated against the gold standard Douglas Bag. The oxygen sensors automatically calibrated before each measurement [24]. Patients were told to come to the hospital at 8.30 in the morning after a 12-h fast and not to exercise the day before the test. After their arrival, they rested lying down for at least 20 min, before a ventilated canopy was placed over their head and shoulders, and they were instructed to breathe freely. The test lasted 20 min with measurements of oxygen consumption at 1-min intervals. REE was calculated from oxygen consumption using a modified Weir equation [25].

### 2.6. Physical Activity and Performance

Spontaneous physical activity was measured using the SenseWear Armband (SWA, BodyMedia, Inc., Pittsburgh, PA, USA). This armband is a multisensory body monitor that contains a two-axis accelerometer, heat flux sensor, galvanic skin response sensor, and skin and near-body temperature sensors. Subjects were told to wear the armband on their upper arm for 24 h/day for 3 days, removing it only for short periods when it might get wet (e.g., while taking a shower or swimming). Metabolic Equivalent Task (MET) expresses the energy expenditure of a physical activity and is defined as the ratio of metabolic rate during a specific physical activity to a reference metabolic rate, set by convention at 3.5 mL O_2_·kg^−1^·min^−1^.

As physical performance tests, we performed the 30-s Sit-to-Stand Chair Test (30” STS), the Six-Minute Walking Test (6′WT), and the Handgrip Strength Test.

The 30′’ STS is a validated test which allows us to assess lower-extremity strength in adults who are over 60 [26,27]. The participant is seated in a chair with his/her arms crossed across their chest and hands resting against their shoulders. The score corresponds to the number of times the person can stand up from a sitting position in 30′’ without the help of their arms. Data were compared with standard values for age and gender.

The 6′WT was performed according to the American Thoracic Society’s guidelines [28]. This test measures the distance a subject is able to walk over six minutes on a hard, flat surface. After resting for 10 min, the subject walks along a path with marked turning points. The patient is allowed to self-pace and rest as needed [28].

The Handgrip Strength Test was performed using a hydraulic hand dynamometer (Jamar, Duluth, MN, USA). The handle of the dynamometer was adjusted to fit comfortably in the subject’s hand with allowance for a good grip. The person was told to place their arm at their side, keeping it away from their body with their elbow bent at a ninety-degree angle. The test was administered on the dominant hand first. An emphasis on “squeeze as hard as you can” was used for maximum effect. We allowed three trials with each hand, right and left hand alternately, with a pause of 10 to 20 s between each trial to avoid excessive fatigue.

The Rapid Assessment of Physical Activity (RAPA) is a brief validated patient-based questionnaire that assesses usual physical activity level in adults over 50 [29]. Patients filled in the questionnaire while waiting for their appointment in the renal outpatient clinic. A RAPA score ≤ 3 corresponds to a sedentary lifestyle or a very low activity level; values ≥ 4 indicate a moderately active to vigorous lifestyle [29].

### 2.7. Diagnosis of Sarcopenia

According to the European Working Group on Sarcopenia in Older People (EWGSOP), sarcopenia can be diagnosed when there is the simultaneous presence of two criteria: low muscle mass and low muscle function (strength or performance) [16]. Among the measurement methods recommended by EWGSOP we used BIA to estimate SMI, as a predictor of muscle mass, and handgrip to measure muscle strength, as described above. Study participants with SMI ≤ 10.75 kg/m^2^ and handgrip strength < 30 kg were defined as sarcopenic [16,30,31,32].

### 2.8. Statistical Analysis

Descriptive analysis has been expressed as mean ± standard deviation (SD) or median and inter-quartile range (IQR) when appropriate. Comparisons between groups were assessed by *t*-test for unpaired data. Comparison between prevalence was evaluated by the chi-square test or Fisher exact test. Spearman’s linear correlation analysis was used to determine associations between selected parameters.

Survival curves were calculated according to Kaplan–Meier and the log-rank test was employed to evaluate differences between curves. Significance was set at 0.05.

Analysis was carried out with SPSS v.25 (SPSS Inc., Chicago, IL, USA).

## 3. Results

### 3.1. Biochemical Data

Table 1 shows the main clinical features of the older and younger senior groups. Residual kidney function was similar. Likewise, no differences were observed in the metabolic profile for BUN, bicarbonate, phosphate, PTH, hematocrit, or potassium, which overall indicates good metabolic control of CKD, in keeping with the choice of analyzing clinically-stable patients. Older senior patients had significantly lower serum albumin (*p* = 0.03). However, serum albumin was in the normal range in both subsets of stable patients.

The difference in Charlson Index is fully explained by age (included in the index itself).

### 3.2. Anthropometry and Body Composition

Table 2 reports the anthropometry and bio-impedance analysis data in the older and younger senior groups. Older senior patients displayed lower BMI, muscle arm circumference (MAC), muscle arm muscular circumference (MAMC), muscle arm muscular area (MAMA), and skeletal muscle mass (SM) than younger patients, whereas no difference in body fluid or fat mass was observed.

### 3.3. Resting Energy Expenditure e Total Energy Requirement

Although resting energy expenditure (REE), assessed by indirect calorimetry, was lower in older seniors (1248 ± 194 vs. 1417 ± 190 Kcal/day, *p* = 0.0004), no significant difference was found between the two groups after normalization for fat-free mass (23.8 ± 5.1 vs. 22.8 ± 5.6 Kcal/kg fat-free mass/day in older seniors and younger seniors, respectively).

Total energy requirements were calculated using indirect calorimetry and average daily METs were lower in the older seniors (1384 ± 222 vs. 1800 ± 265 Kcal/day, *p* < 0.00001), while total energy intake derived from dietary recalls were similar in the two groups (1784 ± 298 Kcal/day vs. 1737 ± 296 Kcal/day, *p* = 0.883). In older seniors, the measured total energy requirement (TEE) was lower than the energy intake estimated from dietary recall, while no difference was found in the younger seniors, after normalization for ideal body weight (younger seniors: 20.0 ± 3.5 vs. 24.8 ± 4.3 Kcal/kg IBW/day; older seniors: 26.3 ± 4.7 vs. 25.3 ± 5.2 Kcal/kg IBW/day).

### 3.4. Estimated Protein and Mineral Intake

Protein intake estimated using the Maroni and Mitch formula was significantly lower in the older seniors and it was in accordance with their lower urine phosphate excretion (Table 3). Energy and nutrient intakes, estimated from the patients’ food journals, were similar in the two groups (Table 3). Protein intakes derived from the food journals were similar to those estimated using the Maroni and Mitch formula in older seniors while it was significantly lower in younger seniors (Table 3).

There was no difference in appetite score as estimated by the Council on Nutrition’s Appetite Questionnaire (CNAQ) (28.7 ± 4.2 vs. 30.4 ± 4.0. *p* = 0.08).

### 3.5. Physical Activity and Performance

Table 2 reports the data regarding physical activity and performance in the study cohort.

Patients’ answers to the RAPA questionnaire showed that 62.5% of the younger seniors and 70% of the older seniors were sedentary/underactive. The average daily METs values were significantly lower in older seniors, confirming a higher prevalence of sedentary/low-activity lifestyle.

Older seniors spent a greater proportion of their time in sedentary pursuits and this was reflected in their lower scores on the 6 min walk test (Table 2).

### 3.6. Sarcopenia Evaluation

A muscle-mass index indicative of sarcopenia was found in 82.5% of older seniors, compared to 62.5% of younger seniors. According to the EWGSOP, criteria, sarcopenia was more prevalent in the older than in the younger group (55 vs. 12.5%, *p* > 0.01).

Sarcopenic patients differed from non-sarcopenic ones in age and in performance on the Six-Minute Walk Test but eGFR, biochemistry, protein, and energy dietary intakes were similar (Table 4). Resting energy expenditure, measured by indirect calorimetry, was not significantly lower in sarcopenic subjects. The measured total energy requirement was similar in sarcopenic and non-sarcopenic patients, both expressed as an absolute value, and after normalization for ideal body weight, and was lower than daily energy intake estimated by dietary recall (Table 4).

Over a mean follow-up of 29 ± 13 months, 23% of sarcopenic and in 11% of non-sarcopenic patients died; albeit clinically relevant, the difference does not reach statistical significance, probably on the account of the relatively short follow-up (*p* = 0.336). No difference was observed in the need for dialysis start, recorded in 18% of sarcopenic and 17% of non-sarcopenic patients (*p* = 0.900).

Furthermore, sarcopenia was not associated with a greater risk for all-cause mortality or ESRD; Figure 1 and Figure 2 report the data observed in the older senior cohort.

## 4. Discussion

Sarcopenia, a term that encompasses reduced muscle mass and function, is an alteration frequently accompanying the physiological process of aging. Sarcopenia is associated with increased morbidity and mortality in several settings, including CKD [34]. Conversely, CKD may be one of the main causes of sarcopenia; since the CKD population is aging, a vicious cycle linking CKD and sarcopenia impairs our patients’ life expectancy [35].

Our study found a high prevalence of sarcopenia in younger and older seniors, as diagnosed from reduced muscle mass and strength assessed, in keeping with the EWGSOP guidelines, using BIA and handgrip test or a simple questionnaire. We used Bioimpedance Analysis to assess skeletal muscle index, according to the European Working Group on Sarcopenia in Older People (EWGSOP) criteria. Our patients were stable, didn’t show over- or under-hydration and fluids were adequately distributed between the intra- and extra-cellular compartments. These data comforted us in the use of BIA to estimate muscle mass. CT and DEXA may be more objective in SMI evaluation but they are expensive, not always available and, for these reasons, not easily applicable in daily clinical practice. In this line, we also chose to use commonly measured laboratory variables, and assessed eGFR by means of a creatinine-based formula, on the account of its higher availability, and of the good correlation observed with cystatin-C-based formula [36].

Nutrient intake, and in particular protein intake, was not associated with sarcopenia in our study: this could be attributable to the fact that patients received personalized dietary counseling, showed good adherence to dietary recommendations (as shown by nPCR) and had a good metabolic profile. This finding may support tailored nutritional therapy and focused counseling for patients and their families. In this context we can conclude that the reduction of muscle mass is mainly attributable to other factors, pointing to the negative influence of physical inactivity.

A collateral finding of potential practical relevance is that in on our study the food journals kept by patients slightly underestimated protein intake compared to PCR, in particular in younger seniors, something that should be kept in mind, given the widespread use of this tool in the clinical practice. This discrepancy may depend on the fact that older seniors tend to have monotonous dietary habits, while younger seniors tend to have more snacks (which are often not reported).

Energy intake was similar in younger and older senior groups, in the presence of a trend towards lower energy requirements in older patients. Overall, our data suggest that recommended energy intake should be revised and probably lowered in elderly patients, considering the high prevalence of sedentary habits and of reduced physical activity, and hence the low total energy requirement. This observation is not novel: Heiwe reported that physical fitness in adults with CKD may reduce even to the point of impairing their ability to perform everyday activities [37]. Age-related mitochondrial dysfunction, reduced insulin sensitivity, and reduced physical endurance are probably not merely the consequence of the aging process but are related to physical inactivity and increased adiposity, in a vicious cycle [9,38].

In this regard, since regular exercise contributes to alleviating age-related mitochondrial dysfunction and improving muscle function, independent from protein intake, our data stress the importance of combining nutritional counseling with physical activity, in keeping with the observation of Thomas and co-workers affirming that a sedentary lifestyle causes a degree of muscle atrophy similar to that found in acute illness or injury [9,38,39].

Our study, the strength of which is the deep nutritional phenotyping of older senior CKD patients that it offers, has several limitations: it is based upon a convenience sample of relatively limited size, and involved only males. This is justified by the fact that the parameters evaluated are strictly related to gender. Furthermore, our study combines a detailed physical assessment of the patients with the usual laboratory markers, each of which displays important limitations. This is particularly true in the case of serum albumin, that is deeply affected by inflammation and hydration. We tried to acknowledge these limitations by selecting patients who were clinically stable and euvolemic, and who did not present, at the moment of the study, any clinical or biochemical sign of inflammation.

Another limit may be the choice of a creatinine-based formula (CKD-EPI) for e-GFR calculation. In fact, the use of cystatin C-based eGFR might be preferable in this setting, avoiding an overestimation eGFR in sarcopenic subjects. Once more our choice was linked to the study design that included commonly tested biochemical data. The issue is still debated, and this topic deserves further specific investigation [36].

In our study sarcopenia was found to be widespread in an elderly male CKD population, and it was related to age and physical inactivity and not to dietary habits or degree of renal-function impairment. Our findings are in keeping with those of Hung and his co-workers, who demonstrated that low-protein diet therapy did not affect nutritional status in CKD patients, and that a low-protein diet was even associated with increased serum albumin in CKD patients over 60 years of age, probably as an effect of careful nutritional counseling [40].

This finding has significant practical relevance since, at least in most European countries, the prevalence of CKD sharply increases in older individuals, and the benefits of slowing the progression of kidney disease and avoiding/delaying dialysis are particularly important in this population. Our study supports that nutritional care including reduced protein intake does not increase the incidence of sarcopenia, a claim frequently made when arguing against the widespread implementation of low-protein diets in the management of CKD patients. At least in our cohort of older CKD patients, sarcopenia was not associated with a greater risk for all-cause mortality or dialysis commencing in the medium term (Figure 1 and Figure 2). This data is not fully in line with the literature that identifies in sarcopenia a clinical element associated with a significant increase in the risk of death. The reasons for this discrepancy may reside in the fact that our patients were clinically stable and on tertiary renal care, and that the observation time was limited; indeed, risk of mortality and ESRD were remarkably low, reflecting the careful selection of the patients and, in such a population, the detection of the effect of the reduced muscle mass may require a longer follow-up.

## 5. Conclusions

Elderly male CKD patients have lower muscle mass, muscle strength, physical capacity, and activity levels, with a high prevalence of sarcopenia; all these alterations increase with age, in the presence of a stable metabolic profile, and of adequate energy and protein intake.

While this finding is not unexpected, in the CKD population the detection of sarcopenia is often considered as a complication of renal insufficiency and attributed to restrictive dietary interventions. In this context, our data suggest that sarcopenia in this subset of stable CKD patients in tertiary care is not a matter of residual renal function or dietary modulation, but is probably determined by older age. This may in turn support the implementation of a dietary interventions aiming to correct metabolic and nutritional abnormalities, in elderly CKD patients.

Moreover, the high prevalence of sarcopenia, found in our study, stresses the importance of a regular monitoring of nutritional and functional status, and suggests systematic integration of exercise and nutritional counseling in this frail population.

## Figures and Tables

**Figure 1 nutrients-10-01951-f001:**
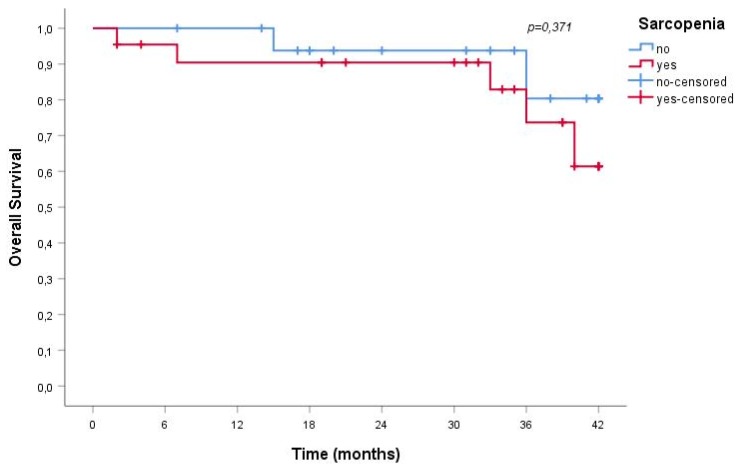
Overall survival rate of sarcopenic and non-sarcopenic older senior CKD patients.

**Figure 2 nutrients-10-01951-f002:**
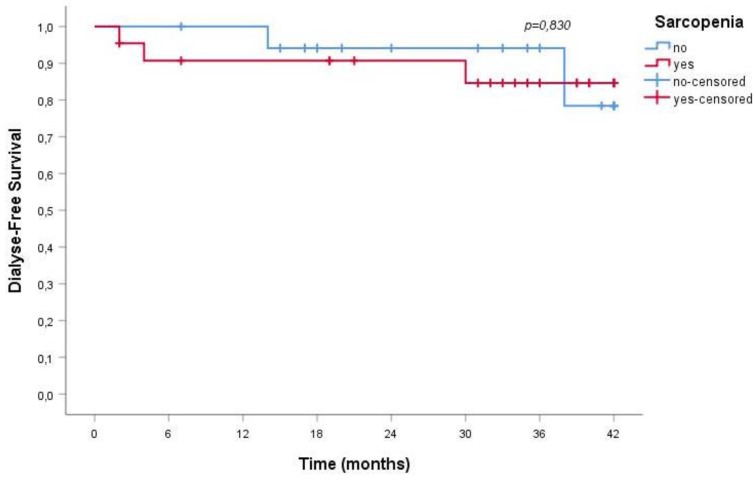
Dialysis-free survival rate of sarcopenic and non-sarcopenic older senior CKD patients.

**Table 1 nutrients-10-01951-t001:** Main characteristics of the chronic kidney disease (CKD) patients studied: older seniors (≥75 years of age) and younger seniors (age 60–74 years of age).

	All Patients (*n* = 80)	Older Seniors (*n* = 40)	Younger Seniors (*n* = 40)	*p* Older vs. Younger Seniors
Age, years	73.7 ± 7.2	79.8 ± 3.3	67.5 ± 4.3	0.0001
Charlson Index (median)	7	7	6	0.01
eGFR, mL/min × 1.73 m^2^	28.3 ± 9.8	30.2 ± 12.6	29.4 ± 9.8	0.31
BUN, mg/dL	42 ± 14	42 ± 11	43 ± 15	0.73
sCreatinine, mg/dL	2.65 ± 1.0	2.7 ± 1.1	2.6 ± 1.0	0.55
sSodium, mEq/L	141 ± 2.2	141 ± 2.5	140 ± 1.7	0.06
sPotassium, mEq/L	4.75 ± 0.5	4.8 ± 0.5	4.7 ± 0.5	0.20
sCalcium, mg/dL	9.25 ± 0.4	9.2 ± 0.5	9.3 ± 0.4	0.87
sPhosphate, mg/dL	3.3 ± 0.6	3.2 ± 0.5	3.4 ± 0.6	0.12
Bicarbonate, mEq/L	24.5 ± 2.9	24.5 ± 2.9	24.5 ± 3.0	0.95
sTotal Protein, g/dL	7.0 ± 0.5	7.1 ± 0.5	7.1 ± 0.5	0.97
sAlbumin, g/dL	4.1 ± 0.4	4.0 ± 0.4	4.2 ± 0.3	0.03
Hemoglobin, g/dL	13.1 ± 1.6	12.9 ± 1.3	13.4 ± 1.8	0.18
Hematocrit, %	39.5 ± 4.4	38.9 ± 3.9	40.1 ± 4.9	0.24

Legend: eGFR, estimated glomerular filtration rate; BUN, blood urea nitrogen.

**Table 2 nutrients-10-01951-t002:** Anthropometry, bio-impedance analysis, physical activity, and performance parameters in younger and older seniors.

	All Patients (*n* = 80)	Older Seniors (*n* = 40)	Younger Seniors (*n* = 40)	*p* Older vs. Younger Seniors
Body weight, kg	80.6 ± 12.6	77.3 ± 10.3	84.0 ± 14.0	0.02
BMI, kg/m^2^	28.2 ± 3.7	27.1 ± 3.3	29.2 ± 3.9	0.012
Waist circ., cm	102.7 ± 13	101 ± 14	104 ± 11.2	0.23
MAMC, cm	25.9 ± 3.2	24.8 ± 2.3	27.0 ± 3.6	0.0014
MAMA, cm^2^	54.2 ± 12.5	49.3 ± 9.2	54.2 ± 13.5	0.0003
Phase angle,	5.0 ± 1.2	4.8 ± 1.3	5.2 ± 1.0	0.18
BCMI, kg/m^2^	9.9 ± 3.0	9.5 ± 3.9	10.2 ± 1.6	0.34
SM, kg	28.0 ± 3.8	26.6 ± 3.5	29.5 ± 3.5	0.001
SMI, kg/m^2^	9.6 ± 1.9	9.0 ± 2.4	9.2 ± 1.9	0.002
RAPA score	2.4 ± 2.2	2.5 ± 1.9	2.4 ± 2.4	0.72
Average daily METS	1.2 ± 0.2	1.1 ± 0.1	1.3 ± 0.2	0.006
PA level > 3METS/min	47.7 ± 50.9	31.7 ± 37.2	62.0 ± 57.7	0.03
Hand grip, kg	32.1 ± 11.9	27.5 ± 6.3	36.8 ± 14.5	0.0004
STS30”, n.rep	10.9 ± 2.8	10.1 ± 3.0	11.7 ± 2.4	0.01
6MWT, m	309 ± 84	282 ± 80	336 ± 80	0.003

Legend: MAMC: middle arm muscle circumference; MAMA: middle arm muscle area; BCMI: body cell mass index; SM: skeletal muscle; SMI: skeletal muscle index; RAPA: Rapid Assessment of Physical Activity; MET: metabolic equivalent; PA: physical activity; STS30”: 30-s Sit-to-Stand Chair Test; 6MWT: Six-Minute Walking Test.

**Table 3 nutrients-10-01951-t003:** Nutrient and energy daily intake estimated from urine collection measurements and from the analysis of patients’ food journals.

	Older Seniors (*n* = 40)	Younger Seniors (*n* = 40)	*p* Older vs. Younger Seniors
**Data from 24-h urinary excretion:**			
PCR, g die	59.1 ± 16.8	74.3 ± 21.6	0.007
nPCR, g/kg IBW/day	0.86 ± 0.25	1.04 ± 0.27	0.015
Phosphate, mg/day	444 ± 222	678 ± 266	0.004
Sodium, mEq/day	131 ± 53	146 ± 66	0.749
**Data from food journals:**			
Energy, Kcal	1663 ± 320	1613 ± 343	0.536
Energy, Kcal/IBW	24.5 ± 5.5	23.2 ± 5.9	0.351
Protein, g	51.5 ± 15.8	52.3 ± 17.9	0.851
Protein, kg/IBW	0.76 ± 0.27	0.74 ± 0.24	0.713
Fats, g	63.8 ± 15.7	62.1 ± 14.4	0.647
Carbohydrates, g	215 ± 47.7	212 ± 59.6	0.768
Phosphate, mg	737 ± 231	731 ± 239	0.914
Potassium, mg	2058 ± 534	2013 ± 567	0.734
Sodium, mg	967 ± 521	853 ± 613	0.407
Fiber, g	16.6 ± 5.9	15.6 ± 4.7	0.466

Legend: IBW, ideal body weight (ideal body weight was estimated according to the Renal Association’s recommendations) [33]; PCR, protein catabolic rate; nPCR, normalized protein catabolic rate.

**Table 4 nutrients-10-01951-t004:** Biochemistry, anthropometry, bio-impedance analysis, physical activity, performance parameters, and nutrient intake derived from an analysis of the food journals kept by sarcopenic and non-sarcopenic older seniors.

	Older Seniors (*n* = 40)		*p* Sarcopenic vs. Non Sarcopenic
	Sarcopenic (*n* = 22)	Non-Sarcopenic (*n* = 18)	
Age, years	81.0 ± 3.4	78.3 ± 2.6	0.01
BMI, kg/m^2^	26.4 ± 3.3	27.9 ± 3.9	0.16
eGFR, mL/min 1.73 m^2^	32.2 ± 15.5	28.0 ± 8.5	0.38
BUN, mg/dL	41 ± 12	42 ± 13	0.81
Bicarbonate, mEq/L	24.7 ± 2.9	24.1 ± 2.8	0.60
sProtein, g/dL	7.0 ± 0.5	7.1 ± 0.5	0.68
sAlbumin, g/dL	4.0 ± 0.4	4.1± 0.2	0.21
Hemoglobin, g/dL	12.8 ± 1.3	13.1 ± 1.4	0.57
Hematocrit, %	39.0 ± 3.9	38.8 ± 4.0	0.90
PCR, g die	55.5 ± 17.8	63.3 ± 15.1	0.25
nPCR, g/kg IBW/day	0.81 ± 0.25	0.93 ± 0.24	0.46
Phase angle,	4.7 ± 1.1	5.0 ± 1.5	0.39
BCMI, kg/m^2^	8.5 ± 1.7	10.8 ± 5.4	0.06
RAPA score	1.9 ± 2.5	2.9 ± 2.1	0.20
Average daily METS	1.16 ± 0.1	1.12 ± 0.1	0.47
STS30”, n.rep	9.4 ± 3.2	11.1 ± 2.7	0.09
6MWT, m	257 ± 82	312 ± 66	0.03
REE, Kcal/day	1196 ± 170	1315 ± 208	0.06
TEE, Kcal/day	1388 ± 237	1379 ± 217	0.93
nTEE, Kcal/kg IBW/day	19.7 ± 4.0	20.0 ± 3.5	0.69
Energy, Kcal/kg IBW/day	25.8 ± 4.1	24.0 ± 4.5	0.27
Protein, g	50.6 ± 15.8	52.8 ± 16.3	0.70
Protein, kg/IBW/day	0.75 ± 0.25	0.78 ± 0.30	0.77

Legend: BCMI: body cell mass index; RAPA: Rapid Assessment of Physical Activity; MET: metabolic equivalent task; STS30”: 30-s Sit-to-Stand Chair Test; 6MWT: Six-Minute Walk Test; REE: resting energy expenditure; TTE: total energy expenditure; nTTE: normalized total energy expenditure; IBW: ideal body weight. BUN: blood urea nitrogen.

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
