# Peer review of "Prevalence and Correlates of Sarcopenia among Elderly CKD Outpatients on Tertiary Care"

_nutrients, 2018, doi:10.3390/nu10121951_

Reviewer 1 Report

Strong study and important findings. Authors should consider adding a de-identified food journal from a selected subject as a supplement to provide evidence of strong patient adherence. Authors should also comment on the general limitations with 24-hour recall and other forms of dietary habit reporting, even in the context of a highly compliant study population.

Authors should comment on the limitation of use of serum albumin used as a malnutrition biomarker. Significance change of serum albumin between older and younger adults could be attributed to inflammation or volume status in older adults.

Author Response

Reviewer 1.

(comments in evidence in yellow)

Strong study and important findings. Authors should consider adding a de-identified food journal from a selected subject as a supplement to provide evidence of strong patient adherence. Authors should also comment on the general limitations with 24-hour recall and other forms of dietary habit reporting, even in the context of a highly compliant study population.

Thank you for your kind comment.

As for the 3 days recall:

The 3-day dietary record was conducted on the basis of widely accepted and applied rules [Hebert I et al, 1997]. Food record analysis represent a valid and simple tool for nutrients and energy estimation but it is well known that potential errors, most often underestimation, may occur. We tried to limit this bias by collecting an interview at the time of delivery of the 3-day food journal, in order to verify, as described in the method section, the sizes of the portion, the type of food and to complete missing data. Recording a food journal for a longer period is probably more appropriate but also at higher risk of underreporting [Dwyer JT, 1994], especially in older people who easily get tired answering too many questions.

Furthermore, we preferred the 3-day record to the food frequency questionnaire since the second is more appropriate for a qualitative rather than quantitative assessment of food habits, and, in this specific case, we felt that quantification of energyand proteins was of particular importance; moreover, the food frequency questionnaire is generally applied in study population rather than small cohorts.[Affret A et al., 2017]

We modified the text accordingly to these comments and we added the following references

Hebert I, Ockene IS, Hurley TG, Luippold R, Well AD, Harmatz MG. Development and testing of a seven-day dietary recall. J Clin Epidemiol. 1997; 50: 925–937.

Dwyer JT. Dietary assessment. In: Shils ME, Olson JA, Shike M, editors. Modern nutrition in health and disease. Philadelphia, USA: Lea & Febiger; 1994, 842-860

Affret A, Wagner S, El Fatouhi D, Dow C, Correia E, Niravong M, Clavel-Chapelon F, De Chefdebien J, Fouque D, Stengel B; CKD-REIN study investigators, Boutron-Ruault MC, Fagherazzi G. Validity and reproducibility of a short food frequency questionnaire among patients with chronic kidney disease. BMC Nephrol. 2017 Sep 15;18(1):297

Authors should comment on the limitation of use of serum albumin used as a malnutrition biomarker. Significance change of serum albumin between older and younger adults could be attributed to inflammation or volume status in older adult

We agree with the reviewer’s concerns, however  in our series we chose stable patients, who showed no signs of over-hydration or lower-hydration, and patients with a pro-inflammatory status, as assessed by CRP , were excluded. It is reported in the patients and methods section, and further commented in the text as follows:

In the results:

However, serum albumin was in the normal range in both subsets of stable patients.

In the limits (discussion):

Furthermore, our study combines a detailed physical assessment of the patients with the usual laboratory markers, each of which displays important limits. This is particularly true in the case of serum albumin, that is deeply affected by inflammation and hydration. We tried to acknowledge these limitations by selecting patients who were clinically stable and euvolemic, and who did not present, at the moment of the study, any clinical or biochemical sign of inflammation.

We would like to thank the reviewer for the keen comments and for the time dedicated at improving our study.

Reviewer 2 Report

In this article, the authors assess the prevalence and correlates of sarcopenia among elderly male patients affected by chronic kidney disease (CKD) followed up in an outpatient nephrology clinic. The topic is of interest. However, there are some critical flaws to be clarified or revised as follows. 

1)  The novelty is poor. Even in patients with CKD, it is well known finding that the incidence of sarcopenia is higher in older seniors than younger seniors.

2)  The stage of CKD depends on serum creatine level. However, the creatine level tends to be lower in patients with sarcopenia due to loss of muscle mass. Namely, CKD is underestimated in sarcopenia patients.

3)  The authors assessed SMI using BIA. However, CKD patients have edema in some degree. BIA would be affected by extracellular fluid including edema and ascites. In such patients, therefore, CT is better to assess SNI than BIA, because CT can evaluate SMI more objectively.

4)  How about the outcomes in patients with sarcopenia and without sarcopenia?

Author Response

Answer to reviewer 2 (comments in evidence in blu)

Comments and Suggestions for Authors

In this article, the authors assess the prevalence and correlates of sarcopenia among elderly male patients affected by chronic kidney disease (CKD) followed up in an outpatient nephrology clinic. The topic is of interest. However, there are some critical flaws to be clarified or revised as follows. 

1. The novelty is poor. Even in patients with CKD, it is well known finding that the incidence of sarcopenia is higher in older seniors than younger seniors.

While we agree that finding increased sarcopenia in older patients is not surprising, we do not fully agree with the comment that the paper is devoid of novelty. In fact, It is well known that the prevalence of sarcopenia increases with age, but in the CKD population, detection of sarcopenia is considered as a complication of renal insufficiency and often attributed to restrictive dietary interventions. 

In this context, our data suggest that sarcopenia in CKD patients on tertiary care is not a matter of residual renal function or dietary modulation, but is probably determined by older age. This may in turn support implementation of dietary intervention targeted to correction of metabolic abnormalities, in elderly CKD patients

Moreover we would like to underline the importance of a regular monitoring of nutritional and functional status with simple, not invasive, not expensive tests easily available in the daily clinical practice.

This could appear a commonplace and trite message, however, we all know the need for wider diffusion of nutritional support in elderly patients.

The following comments were integrated in the conclusions:

In conclusion, elderly male CKD patients have low muscle mass, muscle strength and physical capacity and activity levels, with a high prevalence of sarcopenia; all these alterations increase with age, in the presence of a stable metabolic profile, and with adequate energy intake.

While this finding is not unexpected, in the CKD population, detection of sarcopenia is often considered as a complication of renal insufficiency and attributed to restrictive dietary interventions.  In this context, our data suggest that sarcopenia in this subset of stable CKD patients on tertiary care is not a matter of residual renal function or dietary modulation, but is probably determined by older age. This may in turn support implementation of dietary intervention targeted to correction of metabolic abnormalities, in elderly CKD patients.

Moreover the high prevalence of sarcopenia, found in our study, stresses the importance of a regular monitoring of nutritional and functional status, and suggests systematic integration of exercise and nutritional counseling in this fragile population.

2)  The stage of CKD depends on serum creatine level.

However, the creatine level tends to be lower in patients with sarcopenia due to loss of muscle mass. Namely, CKD is underestimated in sarcopenia patients.

Thank you for pointing out this important problem: we added the following comment in the text

discussion

In this line, we also chose to use commonly measured laboratory variables, and assessed eGFR by means of a creatinine based formula, on the account of its higher availability, and of the good correlation observed with cystatin-C based formulae (36).

Limits

Another limit may be the choice of a creatinine-based formula (CKD-EPI) for e-GFR calculation. In fact, the use of cystatin C-based eGFR might be preferable in this setting, avoiding an overestimation eGFR in sarcopenic subjects. Once more our choice was linked to the study design that included commonly tested biochemical data. The issue is still debated, and this topic deserves further specific investigation (36).

3)  The authors assessed SMI using BIA. However, CKD patients have edema in some degree. BIA would be affected by extracellular fluid including edema and ascites. In such patients, therefore, CT is better to assess SNI than BIA, because CT can evaluate SMI more objectively.

This is an interesting point. The following sentences have been added the discussion:

We used Bioimpedance Analysis to assess skeletal muscle index, according the European Working Group on Sarcopenia in Older People (EWGSOP) criteria. Our patients were stable, didn’t show over- or under-hydration and fluids were adequately distributed between the intra- and extra-cellular compartments. These data comforted us in the use of BIA to estimate muscle mass. CT and DEXA may be more objective in SMI evaluation but they are expensive, not always available and, for these reasons, not easily applicable in the daily clinical practice.

4)  How about the outcomes in patients with sarcopenia and without sarcopenia?

Thank for this important point that led us to add patient and renal survival curves to the text (figures 1 and 2).

The main finding is a lack of difference between sarcopenic and non sarcopenic patients, probably on the account of the selection criteria (stable patients) and of the short duration of observation.

We integrate the text as follows:

Methods:

Descriptive analysis has been expressed as mean ± standard deviation (SD) or median and inter-quartile range (IQR) when appropriate. Comparisons between groups were assessed by t-test for unpaired data. Comparison between prevalence was evaluated by chi-square test or Fisher exact test. Spearman’s linear correlation analysis was used to determine associations between selected parameters.

Survival curves were calculated according to Kaplan-Meier and log-rank test was employed to evaluate differences between curves. Significance was set at 0.05.

Analysis was carried out with SPSS v.25.

Results:

Over a mean follow-up of 29 ± 13 months, 23% of sarcopenic and in 11% of not sarcopenic patients died; albeit clinically relevant, the difference does not reacjh statistical significance, probably on the account of the relatively short follow-up (p=0.336).  No difference was observed in the need for dialysis start, recorded in 18% of sarcopenics and the 17% of not sarcopenics patients (p=0.900).

Furthermore, sarcopenia was not associated with a greater risk for all-causes mortality or  ESRD; figures 1 and 2 report the data observed in the older senior cohort . 

We also added this comment in the discussion:

At least in our cohort of older CKD patients, sarcopenia was not associated with a greater risk for all-causes death or ESRD in the medium term (Figures 1 and 2). This data is not fully in line with the literature that identifies in sarcopenia a clinical element associated with a significant increase in risk of death. The reasons of this discrepancy may reside in the fact that our patients were clinically stable, and the observation time was limited; indeed, risk of mortality and ESRD start were remarkably low, reflecting the careful selection of the patients and, in such a population, detection of the effect of the reduced muscle mass may require a longer follow-up.

In the end, the authors would like to thank a lot the reviewer for the attention dedicated to our study and for the important comments to improve it.

Round  2

Reviewer 2 Report

The authors substantially improved their manuscript according to the reviewers' comments. I have no further questions or remarks.